# Biocontrol and Plant-Growth-Promoting Traits of *Talaromyces apiculatus* and *Clonostachys rosea* Consortium against Ganoderma Basal Stem Rot Disease of Oil Palm

**DOI:** 10.3390/microorganisms8081138

**Published:** 2020-07-28

**Authors:** Yit Kheng Goh, Nurul Fadhilah Marzuki, Tuan Nur Fatihah Tuan Pa, Teik-Khiang Goh, Zeng Seng Kee, You Keng Goh, Mohd Termizi Yusof, Vladimir Vujanovic, Kah Joo Goh

**Affiliations:** 1Advanced Agriecological Research Sdn Bhd, Kota Damansara, Petaling Jaya 47810, Malaysia; nf91fadh@gmail.com (N.F.M.); tuanfatihah@gmail.com (T.N.F.T.P.); tkgohmagic2005@yahoo.com.hk (T.-K.G.); keezs@aarsb.com.my (Z.S.K.); gohyk@aarsb.com.my (Y.K.G.); gohkj@aarsb.com.my (K.J.G.); 2Department of Plant Medicine, National Chiayi University, Chiayi City 60004, Taiwan; 3Department of Microbiology, Faculty of Biotechnology and Biomolecular Sciences, University Putra Malaysia, Serdang 43400, Malaysia; mohdtermizi@upm.edu.my; 4Department of Food and Bioproduct Sciences, University of Saskatchewan, Saskatoon, SK S7N 5A8, Canada; vlv699@mail.usask.ca

**Keywords:** antagonistic fungus, ascomycete, biologic control, mycoparasite, *Ganoderma boninense*, plant growth

## Abstract

Basal stem rot (BSR) disease caused by *Ganoderma boninense* basidiomycetous fungus is the most economically important disease in oil palms in South East Asia. Unfortunately, there is no single most effective control measure available. Tremendous efforts have been directed in incorporation of environmentally friendly biocontrol approaches in minimizing BSR disease. This study investigated the performance of two potential biocontrol agents (BCAs), AAT0115 and AAB0114 strains recovered from oil palm on suppression of BSR in planta, and also assessed their plant-growth-promoting (PGP) performance. ITS rRNA-sequence phylogeny discriminated the two ascomycetous *Talaromyces apiculatus* (Ta) AT0115 and *Clonostachys rosea* (Cr) AAB0114 biocontrol species with PGP characteristics. In vitro studies have demonstrated both Ta and Cr are capable of reducing linear mycelial growth of *G. boninense.* Inoculation of individual Cr and Ta—as well as Cr+Ta consortium—induced a significant increment in leaf area and bole girth of oil-palm seedlings five months post-inoculation (MPI) under nursery conditions. At five months post-inoculation, shoot and root biomass, and nutrient contents (nitrogen, phosphorus, potassium, calcium, magnesium and boron) were significantly higher in Ta-inoculated seedlings compared to control treated with non-Ta-inoculated maize. Chlorophyll and carotenoids contents in rapidly growing oil-palm seedlings challenged with Cr, Ta or a combination of both were not negatively affected. Cr, Ta and Cr+Ta consortium treated seedlings had 4.9–60% BSR disease reduction compared to the untreated control. Co-inoculation of Cr and Ta resulted in increased BSR control efficiencies by 18–26% (compared with Cr only) and 48–55% (compared with Ta only). Collectively, Cr and Ta, either individually or in consortium showed potential as BSR biocontrol agents while also possess PGP traits in oil palm.

## 1. Introduction

Basal Stem Rot (BSR) in oil palm (*Elaeis guineensis* Jacq.) is one of the most devastating oil palm diseases in South East Asia (SEA), particularly in Malaysia and Indonesia—and to a lesser extent in other countries in Africa, Papua New Guinea and Thailand [1]. Ganoderma white rot disease is estimated to inflict economic losses as high as USD 500 million annually in Malaysia [2,3]. *Ganoderma* diseases caused yield losses in oil palms through reducing healthy productive standing palms and decreasing their economic lifespan [4,5]. BSR causes infection and decay of the root system and the lower stem, leading to various common foliar symptoms such as frond fracture, yellowing and smaller young fronds [5,6]. Formation of basidiocarps on the lower stem and in severe cases, toppling of the stem are some of the key signs as well. Among the commonly isolated *Ganoderma* spp. from oil palm with BSR disease, *Ganoderma boninense* Pat. was the most widespread and pathogenic species in oil palm plantations compared to the less pathogenic counterparts, namely *Ganoderma zonatum* Murrill and *Ganoderma miniatocinctum* Steyaert [7].

Different cultural and physical methods, namely soil mounding, surgery, trenching, ploughing and harrowing, fallowing and removal of basidiocarps, as well as chemical means had been investigated. Unfortunately, none has been effective for controlling BSR disease [2]. Lack of reproducible and highly reliable early detection methods further makes the control of *Ganoderma* disease even more difficult [2]. Cultural and physical disease management approaches are generally laborious, costly and time-consuming [2,8,9]. Some of the commonly tested commercial cultural practices, namely sanitation through felling and chipping of diseased palms, as well as deboling of the infected root bole during replanting or at the existing planting—with the aim of minimizing the amount of infected tissues buried or left in the soil—may reduce disease infection rate, if done properly [2,8]. The efficacy of using commercially available fungicides has yet been proven [8] and also hindered by a lack of reliable early detection methods. Furthermore, the potential deleterious effects on the environment due to chemical pesticides are still arguable. Therefore, this has encouraged more research for effective environment-friendly biocontrol approaches [10].

Incorporation of two or more biocontrol candidates—either a combination of fungal–fungal, fungal–bacterial or bacterial–bacterial isolates—in managing various important plant diseases has long been adopted. They also showed better efficacy over the use of a single beneficial microbe [10,11,12,13,14]. Application of fungal consortia of more than one biologic control agents (BCAs) was suggested as a useful approach to minimize the variation of BCAs effectiveness in suppressing diseases. Furthermore, this approach is also capable of maximizing the efficacy of the beneficial plant-growth-promoting (PGP) microbes under diverse, uncontrollable and dynamic environmental conditions [12,15,16,17,18]. Introduction of more than one BCA into the BSR pathosystem of oil palms to manage *G. boninense* is not uncommon. Various previous studies using different microbial consortia—in particular a few endophytic bacterial isolates, arbuscular mycorrhizal fungi and bacteria, *Trichoderma* spp. and bacteria or *Penicillium* spp. and bacteria—has been conducted under controlled conditions and showed the potential of reducing *Ganoderma* disease infection and improving oil-palm growth [19,20,21,22].

In the last decade, only a handful of *Ganoderma*-associated fungi have been isolated and identified, including the ascomycetous *Cladobotryum semicirculare* G.R.W. Arnold, R. Kirschner & Chee J. Chen from *Ganoderma lingzhi* Sheng H. Wu, Y. Cao & Y.C. Dai [23] and *G. boninense* [24], *Scytalidium ganodermophthorum* Kang, Sigler, Y.W. Lee & S.H. Yun parasitizing *Ganoderma lucidum* (Curtis) P. Karst. [25], *S. parasiticum* Y-Kheng Goh, Goh, Y.K. Goh, K.J. Goh from *G. boninense* [26] and a *Phlebiopsis* species-parasitizing *Ganoderma philippii* (Bres. & Henn. Ex Sacc.) Bres. [27]. Some of the *Ganoderma*-associated fungi isolated and described from *Ganoderma applanatum* (Pers.) Pat. and *Ganoderma carnosum* Pat. in 1980s and 1990s were outlined by Helfer [28]. *Cladobotryum semicirculare* was observed to suppress *G. boninense* in in vitro assays [24]. Similarly, *Scytalidium parasiticum*—a potential necrotrophic mycoparasite—was found to reduce oil palm BSR disease incidence and severity in nursery experiments [29]. Agustini and co-workers [27] reported the ability of *Phlebiopsis* species in parasitizing and reducing the viability of *G. philippii* in in vitro assays and they suggested that it is a potential biocontrol agent for *Eucalyptus pellita* F. Muell root rot.

In this study, two *Ganoderma*-associated fungicolous AAT0115 and AAB0114 isolates were isolated from rubber-wood blocks (RWB) pre-colonized with *G. boninense* and oil-palm trunk tissues infected with *G. boninense,* respectively. Based on β-tubulin and ITS sequences, the AAT0115 and AAB0114 were identified as *Talaromyces apiculatus* (Ta) Samson, N. Yilmaz & Frisvad and *Clonostachys rosea* (Preuss) Mussat (Cr), respectively. Both *Talaromyces* and *Clonostachys* are among the most important fungal chitinolytic antagonists used as BCAs, with some PGP activities in agriculture crops [30,31]. Hence, we studied Ta-AAT0115 and Cr-AAB0114 abilities to suppress *Ganoderma boninense* growth, reduce *Ganoderma* disease incidence and severity in a nursery oil palm experiment, either applied individually or in combination. Furthermore, the capabilities of Cr or Ta—either applied as a consortium or individually for improving oil-palm growth, nutrient status and chlorophyll contents—were also investigated under nursery environments.

## 2. Materials and Methods

### 2.1. Fungal Isolation and Maintenance

Rubber-wood blocks (RWB) with the size of 3 cm × 3 cm × 3 cm were prepared and artificially inoculated with *Ganoderma boninense* G14 according to Kok et al. [32]. Mesh bags with inoculated RWB were transferred and buried in a coastal-soil, Blenheim series (*Typic Quartzipsamments*) (obtained from Blenheim Estate at coordinates: 3°55′39.40″ N; 100°48′48.58″ E), and further incubated for an additional month in the dark at 24 °C. Mycelial masses of AAT0115 proliferated on RWB pre-colonized with *G. boninense* were collected using flame-sterilized needle under a dissecting microscope and plated onto malt extract agar (MEA) (Difco, Becton Dickinson Diagnostics, Sparks, Maryland, USA) supplemented with antibiotic (100 μg/L of streptomycin sulfate and 12 mg/L kanamycin sulfate) (Sigma-Aldrich, St. Louis, Missouri, USA). AAB0114 isolate was isolated from oil-palm trunk tissues infected by *G. boninense*: Diseased oil-palm trunk tissues were collected from Stothard Estate (coordinates: 5°35′43.37″ N; 100°42′58.41″ E) and incubated on *Ganoderma*-selective medium (GSM) [33,34] for a month in the dark at 24 °C. AAB0114 proliferated on the diseased tissues on GSM, and was transferred onto MEA supplemented with antibiotics. Pure cultures of AAT0115 (Ta) and AAB0114 (Cr) were maintained on MEA prior to DNA extraction, in vitro and in planta (nursery) experiments. AAT0115 and AAB0114 inoculants used for in planta (nursery) experiments were prepared according to Goh et al. [29]: One cm plugs of AAT0115 or AAB0114 mycelial culture were inoculated onto 50 g of sterilized maize and incubated for 4 weeks at 24 °C in the dark. Maize was presoaked twice with double-distilled water, air-dried overnight, and autoclaved at 121 °C for 20 min prior to inoculation. The inoculated 4-week-old maize seeds were air-dried under laminar flow overnight, prior to blending into powder using a Waring^®^ laboratory blender.

### 2.2. Molecular and Phylogenetic Analyses of AAT0115 and AAB0114

AAT0115 and AAB0114 were cultured on MEA and incubated at 24 °C for a week prior to DNA extraction. Genomic DNA was extracted with the DNeasy plant mini kit (Qiagen, Hilden, Germany). ITS (Internal transcribed spacer) fragments were amplified using ITS1 F/ITS4 primer set (ITS1 F—5′-CTTGGTCATTTAGAGGAAGTAA-3′ and ITS4—5′-TCCTCCGCTTATTGATATGC-3′) [35] for both AAT0115 and AAB0114 isolates. Polymerase chain reaction (PCR) was performed according to Vujanovic and Goh [36] with Qiagen Top Taq PCR core kits (Qiagen, Hilden, Germany) and PCR conditions for ITS were outlined previously [37]. Beta-tubulin (*BenA*) region (one of the commonly used regions for identification of *Talaromyces/Penicillium* genera) [38] was amplified with primer set Bt2a (5′-GGTAACCAAATCGGTGCTGCTTTC-3′) and Bt2b (5′-ACCCTCAGTGTAGTGACCCTTGGC-3′) and similar PCR conditions were adopted (Glass and Donaldson 1995) for AAT0115 isolate only. All PCR products were purified using QIAquick PCR purification kit (Qiagen, Hilden, Germany) and purified PCR products were sent to Genomics (Taipei, Taiwan) for sequencing. The ITS sequences obtained from AAT0115 and AAB0114 were submitted to GenBank under the accession Nos.: KY421922 and MG754415, respectively. Furthermore, beta-tubulin sequence for AAT0115 was deposited as KY421924. Sequences of ITS and beta-tubulin gene from the current study and a few previous studies were retrieved from GenBank and were aligned using Clustal W [39] and edited in MEGA version 6 [40]. Maximum likelihood (ML) analyses were carried out using MEGA version 6 software with the following settings: Tamura–Neil model; bootstrap analyses of 1000 repetitions; and nearest-neighbor-interchange ML heuristic method [40]. Phylogenetic trees were generated with sequences illustrating bootstrap values higher than 50%. Trees based on ITS sequences for AAT0115 and AAB0114 were rooted with *Trichocoma paradoxa* CBS788.83 (JN899398) and *Isaria japonica* BCC2821 (EU828662) sequences, respectively. Furthermore, tree based on beta-tubulin sequences for AAT0115 was rooted with *T. paradoxa* CBS788.83 (KF984556).

### 2.3. Dual-Culture Bioassays

Four *G. boninense* isolates, namely, G10, G14, G8 and G12 with different pathogenicity levels (determined in previous study) [32] were selected to inoculate with Ta and Cr in dual-culture assays to assess their capability in suppressing the growth of four selected *G. boninense* isolates on MEA. Ta or Cr mycelial plugs (10-mm diameter) were excised using a 10-mm diameter cork-borer from actively growing colonies and placed 2 cm from each of the *G. boninense* isolates in Petri dishes containing MEA. Linear mycelial growths (mm) of the four *G. boninense* isolates from the respective co-culture pairings in five replications were measured and recorded on alternate day for 2 weeks.

### 2.4. Plant-Growth-Promoting Analysis and Determination of Leaf Chlorophyll Contents

Germinated oil-palm seeds (Dumpy Yangambi Avros DxP) (purchased from Applied Agricultural Resources Sdn. Bhd, Selangor, Malaysia) were planted, maintained and fertilized (drenched weekly using Bayfolan^®^ foliar fertilizer with N:P:K—11:8:6 at the rate recommended by manufacturer) as described by Kok et al. [32] for 2 months. Plants were watered twice per day (approximately 45 min per session). The experiment was carried out under a 50% polyethylene shading net. Two-month-old seedlings with similar uniformity were selected and transferred to 38 × 51 cm black polythene bags filled with Bungor series soil (*Typic paleudult*) (fine sandy clay) (soil chemical parameters—Appendix A) to determine the effects of AAT0115 (Ta) and AAB0114 (Cr)—applied either individually or in combination—on the growth of oil-palm seedlings and leaf chlorophyll contents. Descriptions of all the treatments and controls adopted in this experiment are outlined in Table 1. During transplanting, two-month-old seedlings were inoculated with 50 g of blended Cr (AAB0114) or Ta (AAT0115) inoculant at the base of the stem bole and the roots, for Cr50 or Ta50 treatments. For the treatment with combination of Cr and Ta inoculant at 25 g rates, twenty-five grams of blended Cr and Ta inoculants (total of 50 g of inoculants), respectively, were applied for treatment of Cr25+Ta25. In control 1, the seedlings were inoculated with 50 g of blended uninoculated maize inoculum (Ctrl 1), whereas, in control 2, there was no application of inoculated or uninoculated maize (Ctrl 2). Three treatments and two controls with 6 replicates for each treatment in randomized complete block design (RCBD) were used in the nursery study. Height (cm) (H), leaf area (cm^2^) (LA) and girth of the bole (diameter) (cm) (G) of the seedlings were measured, calculated and recorded at 1, 3 and 5 months post-inoculation (MPI), following the nondestructive techniques reported previously [41,42,43,44]. Leaf chlorophyll and carotenoid contents were extracted and assayed according to the procedures outlined by Sim et al. [45], with some minor modifications: Six leaf discs (leaf tip, middle and base—2 from each respective regions) were collected at 5 MPI using a conventional paper puncher and weighed prior to placing in 500 μL 80% acetone before storing in the dark at 4 °C overnight. Extraction was repeated on the same leaves and the supernatants were pooled and used for analysis. The extractants were measured using UV-Vis Spectrophotometer at 470 nm, 647 nm and 663 nm. Chlorophyll a, chlorophyll b and carotenoid contents were calculated using the formulae proposed by Lichtenthaler and Buschmann [46]: (a) chlorophyll a (μg/mL) = 12.25 A_663_ − 2.79 A_647_; (b) chlorophyll b (μg/mL) = 21.50 A_647_ − 5.10 A_663_; and (c) carotenoid (μg/mL) = (1000 A_470_ − 1.82 chlorophyll a − 85.02 chlorophyll (b)/198.

### 2.5. Plant Nutrient Analyses

Above-ground (shoot: leaf, rachis and bole) and below-ground (root) samples of all seedlings were harvested at 5 MPI and fresh weights were recorded for all three treatments and two controls, namely +Cr50, +Ta50, +Cr25+Ta25, control 1 (uninoculated blended maize) and control 2 (with only seedling) (Table 1). Shoot and root samples were rinsed and washed with sterilized distilled water. Shoot (leaf, rachis and bole) and root samples were oven-dried at 70 °C until constant weight was achieved, and the dry weights were recorded. Oven-dried shoot and root samples were ground and sent to Applied Agricultural Resources Sdn Bhd (AAR) Chemistry Analytical Laboratory (Selangor, Malaysia) for macro- and micronutrient analysis. The shoot and root samples were analyzed for nitrogen (N), phosphorus (P), potassium (K), calcium (Ca), magnesium (Mg) and boron (B) according to methods adopted by Sharifuddin et al. [47].

### 2.6. Nursery Biocontrol Experiment

Two-month-old oil-palm seedlings (Dumpy Yangambi Avros DxP) were used to assess the efficacy of Ta and Cr, either applied individually or in combination, in reducing *G. boninense* (G10 isolate) disease incidence (DI) and severity in the nursery. The oil-palm seedlings were prepared by growing germinated seeds for two months in 15 × 21 cm black polythene bags filled with cocopeat prior to artificial inoculation with *Ganoderma*-inoculated RWB and transplanted to 38 × 51 cm polyethene bags filled with Bungor series soil. Plants were watered twice per day (approximately 45 min per session) and experiment was carried out under 50% polyethylene shading net. Artificial inoculation of RWB at the size of 6 cm × 6 cm × 6 cm with *G. boninense* (G10 isolate) was prepared according to [32]. Similar treatments and controls adopted for plant-growth-promoting study (Table 1) were used in the biocontrol experiment with one additional treatment and some minor modifications. During transplanting, *Ganoderma*-inoculated (G10 + Cr50, G10 + Ta50 and G10 + Cr25 + Ta25) or uninoculated RWB (-G10–Cr–Ta) were placed below the 2-month-old seedlings. Inoculum of BCAs was applied together with the *Ganoderma* inoculum during the transplanting. Fifty grams of blended Cr or Ta inoculant was applied to the base of the stem bole and the roots for G10 + Cr50 and G10 + Ta50 treatments. For the treatment with both Cr and Ta inoculants at 25-g rates, twenty-five grams of blended Cr and Ta inoculants (total of 50 g of inoculants), respectively, were applied for G10 + Cr25 + Ta25-treatment. In the control, seedlings were challenged with uninoculated RWB and 50 g of blended uninoculated maize inoculum (-G10–Cr–Ta). Four treatments and one control with 10 replicates for each treatment in RCBD were used in the nursery study. Observations and scoring for the appearance of *Ganoderma* infection symptoms or signs were recorded at monthly intervals up to 5 MPI.

DI and disease severity index (DSI) were calculated based on the formulae reported earlier by Campbell and Madden [48] and Sapak et al. [49]. Formula for DI = (number of seedlings identified as diseased / number of seedlings per treatment set) × 100%. DSI was calculated using the following formula: DSI = (number of seedlings in the rating × rating number or disease class value) / (total number of seedlings assessed × highest rating or disease class value). Disease classes adopted for DSI calculation were described in [32]. Percent of disease reduction was determined based on the formula given by [49]. The area under the disease progress curve (AUDPC) based on DSI from 1 to 5 MPI was calculated with the formula proposed by Sinko and Piepho [50].

### 2.7. Statistical Analyses

Difference in means for linear mycelial growth of the selected *G. boninense* isolates inoculated with or without AAT0115 and AAB0114 on MEA, as well as means separation of three different vegetative growth measurement (VGM) parameters on 3- and 5-MPI for the respective treatments (without *Ganoderma* inoculum) were analyzed using ANOVA–Fisher’s test with SPSS version 16.0 (SPSS, Inc., Chicago, IL, USA). Means for the major nutrients analyzed in shoot and root for the five treatments without *G. boninense*—as well as means of DSI for oil-palm seedlings challenged with *G. boninense* and inoculation of biocontrol agent(s) at 3- and 5-MPI—were not normally distributed under Shapiro–Wilk test and were transformed where necessary using arcsine–square root transformation prior to ANOVA–Fisher’s test (Minitab version16, Minitab, Inc., State College, PA, USA). Boxplots were generated with the R ggplot2 package [51]. Principal component analyses (PCA) were performed using vegetative growth parameters (H, LA and G) recorded at 3- and 5-MPI, shoot and root major nutrient analyses (N, P, K, Ca, Mg and B), as well as shoot and root dry weights extracted from plant-growth-promoting experiment (recorded for three treatments and two controls: Cr50, Ta50, Cr25+Ta25, Ctrl 1 and Ctrl 2) with RStudio [52].

## 3. Results

### 3.1. Molecular Identification of AAT0115 and AAB0114 Isolates

Two ascomycetous isolates, namely AAT0115 (Figure 1A) and AAB0114 (Figure 1B), were recovered from *G. boninense*-colonized RWB (as a bait) and oil-palm trunk tissues infected by *G. boninense*, respectively. Both AAT0115 and AAB0114 were constantly isolated and cultured from *G. boninense*-colonized RWB and *G. boninense*-infected palm trunk tissues, respectively, over 2 to 3 isolation attempts to isolate *G. boninense*. Based on the NCBI nucleotide database BLAST search using ITS sequences, the closest hits for AAT0115 and AAB0114 were *Talaromyces aculeatus* (100% percent identity toMH316150 with 100% query coverage) and *Clonostachys rosea* (99.6% percent identity to MK203790 with 100% query coverage), respectively. Sequence alignments and ML analysis at 1000× bootstrap repetitions for the ITS sequences placed AAT0115 close together with *T. apiculatus* but separated from *T. aculeatus* (Figure 2A) and AAB0114 within the clade as two other *C. rosea* isolates (Figure 2B). AAT0115 had biverticilliate conidiophores and globose rough to echinulate conidia (Appendix A), with no ascomata. Whereas, AAB0114 had relatively complex conidiophores with mainly biverticilliate, hypaline and smooth primary conidiophores and penicillate-like secondary conidiophores (Appendix A). AAB0114 also produced hyaline, distally broadly rounded and slightly curved conidia. In order to differentiate AAT0115 from *T. aculeatus*, beta-tubulin sequence was obtained and sequence alignment and ML analysis at 1000x bootstrap repetitions with beta-tubulin sequences placed AAT0115 close together with *T. apiculatus* (Figure 3).

### 3.2. In Vitro Ganoderma Boninense Growth Suppression

*Talaromyces apiculatus* AAT0115 (Ta) and *Clonostachys rosea* AAB0114 (Cr) significantly suppressed the growth of all four pathogenic *G. boninense* isolates in dual-culture assays regardless of the pathogenicity status of *G. boninense* (Table 2). *Ganoderma* linear growth suppression by Cr at 2 weeks post-inoculation (WPI) was comparable to Ta. However, Cr was more aggressive at the later time points, being able to degrade *G. boninense* mycelia and hyphae or cause “clearing zone” on the dual-culture assays approximately after 3–4 WPI (Appendix A). Initiation of chlamydospore-like structures (Appendix A) and cell degradation (Appendix A) of *G. boninense* mycelia were observed at the interaction zone with Cr on dual-culture and water agar (WA) slide assays. Barrage lines and mycelia at the interaction zone were degraded and the clearing zone formed at 4 WPI (Appendix A). In contrast, barrage line at the confrontation zone between Ta and *G. boninense* was denser at 4 WPI (Appendix A). In WA slide assay, Ta produced various biotrophic contact structures, namely haustoria-like and peg-like organs, on *G. boninense* cells (Appendix A). Internal penetration and growth within *G. boninense* mycelia by Ta were also observed (Appendix A). Based on preliminary observation, both Cr and Ta could potentially be the antagonist and biotrophic parasitic fungi, respectively.

### 3.3. Plant-Growth-Promoting Abilities of Clonostachys rosea and Talaromyces apiculatus

Inoculation of Cr and Ta individually or in consortium significantly improved oil-palm seedling leaf area, girth diameter of the bole and height compared to the controls (with or without uninoculated maize) at 5 months post-inoculation (MPI) (Figure 4A–C). Furthermore, the height of oil-palm seedlings inoculated with only Ta was significantly taller compared to the controls (Ctrl 1 and Ctrl 2) (Figure 4B). Similarly, root and shoot dry weights of the seedlings inoculated with only Ta were significantly higher compared to the controls at 5 MPI (Figure 5). At the end of the experiment (5 MPI), root and shoot samples were analyzed for the pertinent major and minor nutrients. Treatments with either Cr, Ta or both showed better root and shoot nutrient levels, in particular N, P, K, Ca, Mg and B, compared to control 1 (uninoculated maize only)(Figure 6A–F). Treatment with only Ta demonstrated significantly greater root and shoot nutrients compared to the controls as well (Figure 6A–F). PCA plots, based on first three components, were able to explain approximately 89% of the selected variables (Figure 7A,B). Early plant vegetative growth parameters (at 3-MPI) were positively associated with Cr inoculation (Figure 7B) while Ta inoculation illustrated better correlation with plant nutrient status (Figure 7A,B). Consortium of Cr and Ta improved both plant growth and plant nutrient status (Figure 7B). All the treatments inclusive of the controls had similar chlorophyll (a and b) and carotenoids contents (Figure 8).

### 3.4. Efficacy of Clonostachys rosea and Talaromyces apiculatus in controlling Ganoderma Basal Stem Rot Disease of Oil Palm

In nursery experiment, combinations of Cr and Ta (at the doses of 25 g Cr + 25 g Ta and 50 g Cr + 50 g Ta) reduced disease incidence (DI) to 50–60% from 100% (control) (Table 3). Cr and Ta consortium also contributed to 52% to 60% disease reduction (DR) (derived from area under disease progression curve based on disease severity indices) compared to the treatments with either Cr or Ta alone. Increased in the inoculum quantity of Cr and Ta (consortium) gave better *Ganoderma* disease reduction in oil palms. Application of Cr and Ta consortium at higher rate (50 g Cr and 50 g Ta) significantly reduced both disease severity index (DSI) and area under disease progres curve (AUDPC) (Table 3).

## 4. Discussion

A combination of two fungicolous *T. apiculatus* and *C. rosea* (Ascomycetes) demonstrated biocontrol activity against *G. boninense* (Basidiomycete) BSR disease and also plant-growth-promoting (PGP) effect on oil-palm seedlings. Using RWB pre-inoculated with *G. boninense* (as bait) technique and *G. boninense*-infected oil-palm trunk tissues collected from the field, *Talaromyces apiculatus* AAT0115 and *Clonostachys rosea* AAB0114, respectively, were isolated. Microscopic observation also showed that *T. apiculatus* AAT0115 produced biverticilliate conidiophores and globose rough to echinulate conidia (Appendix A). There was no sexual fruiting bodies observed. These morphologic features were also reported in *T. aculeatus* and *Talaromyces verruculosus* species [38]. Further, *C. rosea* AAB0114 produced biverticilliate or verticillium-like primary conidiophores, penicillate-like secondary conidiophores and slightly curved and distally broadly rounded conidia (Appendix A). These morphologic characteristics produced by *C. rosea* AAB0114 were also described previously by Schroers et al. [53].

In vitro bioassays demonstrated that both *T. apiculatus* and *C. rosea* were capable of suppressing four tested pathogenic *G. boninense* isolates (Table 2). Despite lower *G. boninense* growth suppression at two weeks post-inoculation, Cr began to degrade *G. boninense* mycelia at 3–4WPI (Appendix A). This phenomenon has contributed to the induction of chlamydospore-like resting structures, cell degradation or lysing and vacuolation in *G. boninense* mycelia (Appendix A). Formation of chlamydospore-like survival resting organs, swollen structures and degradation or lysing of host mycelial cells were some of the common effects due to antibiosis mechanism (caused by highly potent lytic enzymes or secondary metabolites) at the confrontation points with bacterial or fungal antagonistic biocontrol agents [54,55,56]. Antagonistic *C. rosea* was also reported as potential biocontrol agent for other diseases [57] as it carries in its genome gene clusters which encode for the synthesis of antifungal polyketide (e.g., clonorosein A–D by *C. rosea*) that is pivotal for fungal–fungal antagonism [58]. Furthermore, *C. rosea* also upregulated *cr-nag1* (highly similar to n-acetyl-beta-d-glucosaminidase) gene during confrontation with *Fusarium culmorum* and indicated the production of chitin-hydrolyzing agents [59]. Upregulation of genes responsible for production of antifungal and cell wall degrading agents may play imperative role in suppressing *G. boninense* growth. On the other hand, *T. apiculatus* produced a few common haustoria- and peg-like biotrophic contact organs, as well as established intracellular penetration into *G. boninense* host (Appendix A). These contact structures were also reported in other biotrophic mycoparasites of *Fusarium* species [36,60,61]. To-date, there is little information related to the potential biotrophic mycoparasitic association between Ta and *G. boninense*. Two other *Talaromyces* species, in particular *T. pinophilus* and *T. flavus* had been reported previously as potential mycoparasite candidates [54]. Furthermore, a few other *Talaromyces* species, namely *T. helices, T. indigoticus, T. rotundus* and *T. wortmannii* showed mycelial growth inhibitory effects toward various fungal pathogens (*Fusarium, Colletotrichum, Phytophthora, Rhizoctonia* and *Sclerotium* species) [62]. In a few previous studies, *T. apiculatus* was reported to produce a list of fungal extrolites (macrocyclic polyactones) in which five of the compounds, namely NG-012, BK-223B, BK-223C, 15G256β and 15G256α-2 were found to have antifungal activity [38,63,64,65].

In this study, leaf area, bole-girth diameter and root dry weight of the oil-palm seedlings were improved through inoculation of Cr, Ta or consortium compared to the uninoculated controls (Figure 4, Figure 5 and Appendix A). To the best of our knowledge, there is very little information related to the plant-growth-promoting effect of *T. apiculatus* in oil palm. This is the first study on the potential plant-growth-promoting effects by Ta or in combination with Cr in oil palm. In a few studies, *Talaromyces* species, namely *T. wortmannii, T. flavus* and *T. pinophilus* promoted the growth of cabbage, cotton and potato and rice seedlings, respectively [31,66,67]. PCA plots also illustrated the positive relationship between Cr inoculation and early oil-palm growth (at 3-MPI) (Figure 7B). Growth-promoting effects by *C. rosea* AAB0114 in oil-palm seedlings were in accordance with a few previous studies. *Clonostachys rosea* IK726 isolate inoculation contributed to an increase of 5–15% in barley shoot dry weight [68]. Endophytic *C. rosea* strain 88–710 was observed to improve the growth of flower plants and cucumbers as well [69]. In our study, oil-palm seedlings inoculated with *T. apiculatus* demonstrated significant higher shoot and root N, P, K, Ca, Mg and B compared to the control with uninoculated maize (Figure 6). Inoculation of Ta was positively correlated with better plant nutrient status (Figure 7A,B). This agrees with a previous study that showed *T. pinophilus* inoculation enhanced nutrients uptake, in particular N, P and K, by pomegranate (*Punica granatum* L.) [70]. Phosphate solubilizing *T. pinophilus* was also found to improve soil availability of P by 1.4–15.4% and increase P content in soybean plants by 1.3–8.9% [71]. A more detailed study on nutrient-solubilizing or -uptake capabilities by Cr and Ta could shed some lights on the improvement of nutrient contents recorded. Increase in both vegetative growth parameters did not negatively affect chlorophyll (chlorophyll a and b) and carotenoids contents. This could also be due to improvement in the leaf nutrient status and nutrient uptake.

In the nursery experiment, the Cr and Ta consortium at highest tested rate (50 g Cr + 50 g Ta) demonstrated greater *Ganoderma* disease reduction compared to treatments with single inoculation—either Cr or Ta (Table 3). Higher dosage of Cr and Ta consortium should also be evaluated in the future studies to determine whether the efficacy of disease controlling will be maintained as observed in the treatment with consortium of 50 g Cr + 50 g Ta. The enhanced control could be the cumulative results of a combination of antagonistic/destructive parasitism by Cr (distance active) and biotrophic parasitism by Ta (intracellular) mechanisms, improvement in plant growth and health, as well as better nutrient status. Furthermore, improved disease control by Cr and Ta could be also due to induction of defense-related mechanisms and genes in the plants toward *G. boninense* infection. Inoculation of *C. rosea* biocontrol agent was reported to induce disease resistance toward *Botrytis cinerea* and *Fusarium oxysporum*, respectively, in tomato plants [72,73]. In a separate study with *Talaromyces wortmannii*, this fungus was found to produce *β*-caryophyllene (terpenoid-like volatiles) with plant-growth-promoting and also disease resistance induction properties [66]. Additionally, combination of multiple potential biologic control agents (BCAs) with different mechanisms were reported to enhance the efficacy of the BCAs, minimize the inconsistency of biocontrol efficacy and improve durability of biocontrol [12,17,74]. Moreover, the ability of fungi in ameliorating a series of biotic and abiotic elements, predominantly through augmentation of plant’s nutrient uptake, could also be one of the important mechanisms adopted by biocontrol agents in ensuring a better palm health for better disease control [53,75]. In order to determine the full potential of both Cr and Ta in promoting growth of oil palms and also *Ganoderma* disease suppression, it is essential to carry out similar experiments in field tests. In addition, it is also pivotal to combine formulation tests with field experiments to optimize the amount of BCAs inoculum and frequency of application required in the perennial oil palm ecosystem for the BCAs to reach their full potential in reducing *Ganoderma* BSR disease.

## 5. Conclusions

Together with development of highly precise early *Ganoderma* BSR detection techniques and availability of potential disease tolerant/resistant oil palm materials, biologic control represents a publicly acceptable, eco-friendly or green approach for reducing *Ganoderma* BSR disease in oil palm. Under in vitro conditions, both *C. rosea* AAB0114 and *T. apiculatus* AAT0115 were capable of suppressing the growth of four selected pathogenic *G. boninense* isolates. Furthermore, a combination/consortium of aggressive antagonistic *C. rosea* and intracellular biotrophic mycoparasitic *T. apiculatus* BCAs isolated from *Ganoderma*-infected substrates could improve plant growth and nutrients status and reduce the incidence and severity of oil palm *Ganoderma* BSR disease. Although, reduction in *Ganoderma* BSR disease scorings was not high, *Talaromyces apiculatus* if applied singly could enhance significantly greater plant growth and biomass. Application of the *C. rosea* and *T. apiculatus* consortium or only *T. apiculatus* could be the potential mycofungicide and biofertilizer products, respectively, in the future for improving oil-palm growth and reducing *Ganoderma* BSR disease.

## Figures and Tables

**Figure 1 microorganisms-08-01138-f001:**
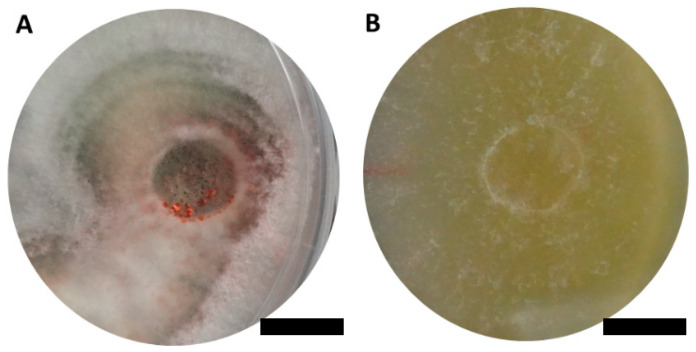
Colony appearances of (**A**) *Talaromyces apiculatus* (Ta) AAT0115 and (**B**) *Clonostachys rosea* (Cr) AAB0114 on malt extract agar incubated for 7 days. Scale bar: 10 mm.

**Figure 2 microorganisms-08-01138-f002:**
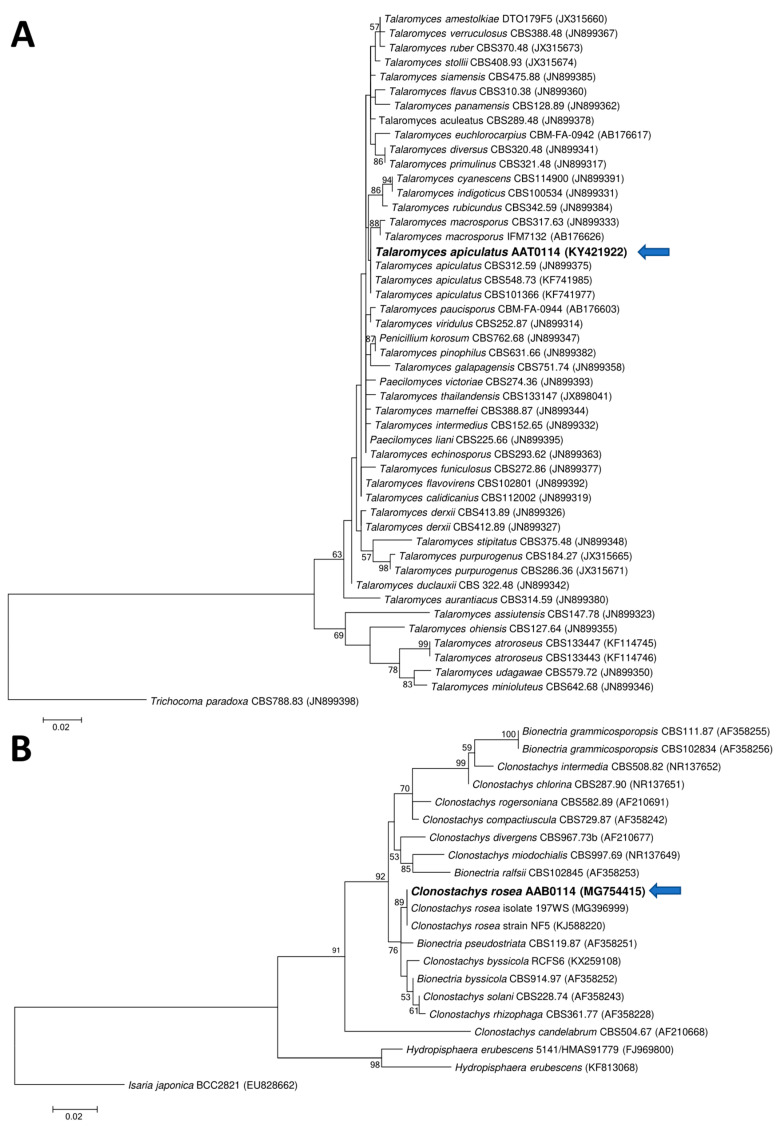
Phylogenetic relationship of (**A**) AAT0115 or (**B**) AAB0114 and their closely related isolates/species based on the internal transcribed spacer (ITS) rDNA sequences and positions of currently studied ascomycetes (in bold) (arrow). Two fungal taxa—namely *Trichocoma paradoxa* CBS788.83 (JN899398) and *Isaria japonica* BCC2821 (EU828662)—were used as the outgroups for the respective phylogenetic trees generated through maximum likelihood approach. Bootstrap values of ≥50% from 1000 bootstrap repetitions are shown for the corresponding branches. Scale bars indicate estimation of 0.02 substitutions per nucleotide position for the branch lengths.

**Figure 3 microorganisms-08-01138-f003:**
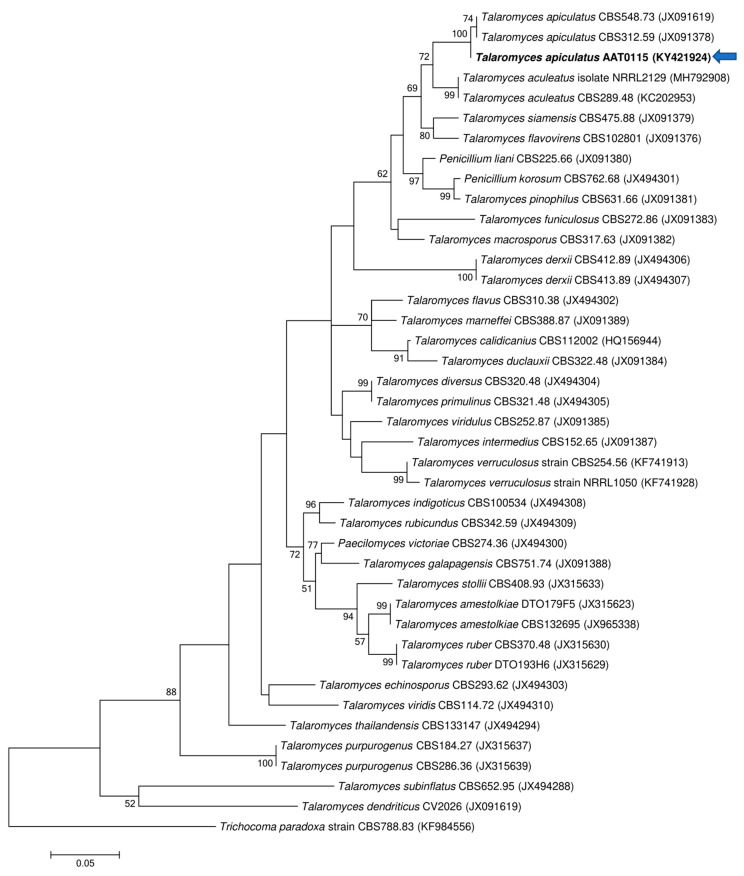
Phylogenetic relationship of AAT0115 and its closely related species based on beta-tubulin (*BenA*) sequences and position of the AAT0115 (in bold) (arrow). *Trichocoma paradoxa* CBS788.83 (KF984556) was used as the outgroup for the phylogenetic tree generated through maximum likelihood approach. Bootstrap values ≥50% from 1000 bootstrap repetitions are shown for the corresponding branches. Scale bars indicate estimation of 0.05 substitutions per nucleotide position for the branch lengths.

**Figure 4 microorganisms-08-01138-f004:**
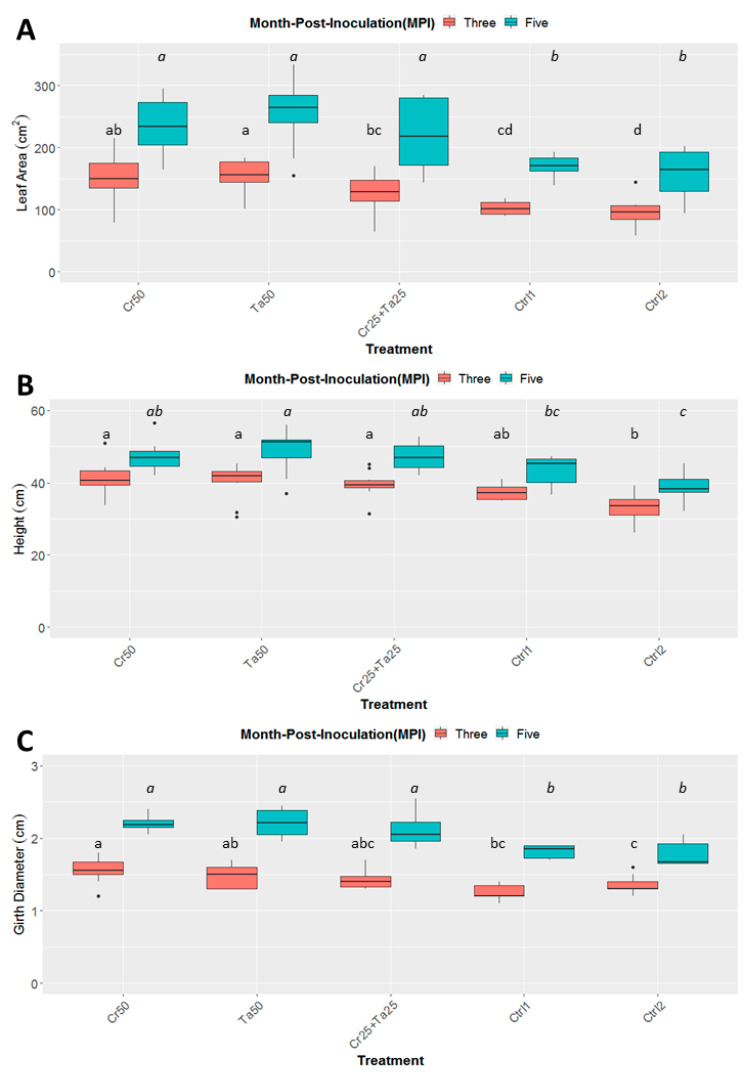
Effects of *Ta* and *Cr*, either applied individually or in consortium on the development and growth of oil-palm seedlings, namely leaf area (**A**), height (**B**), and diameter of the bole girth (**C**), 3 and 5 months post-inoculation (MPI). Treatments abbreviations: Cr50—inoculated with 50 g of *C. rosea* AAB0114 (*Cr*); Ta50—inoculated with 50 g of *T. apiculatus* AAT0115 (*Ta*); Cr25 + Ta25—inoculated with 25 g of *Cr* and 25 g of *Ta*; control 1—only the uninoculated blended maize was applied; and control 2—without the uninoculated blended maize. Means of the three different vegetative growth parameters (VGM) at two separate recording time points (3 and 5 MPI) were analyzed separately. Means with the same letters for the all five treatments within 3 or 5 MPI for the respective VGM parameters were not significantly different with ANOVA–Fisher at *p* = 0.05. Numbers presented in the figures are untransformed means.

**Figure 5 microorganisms-08-01138-f005:**
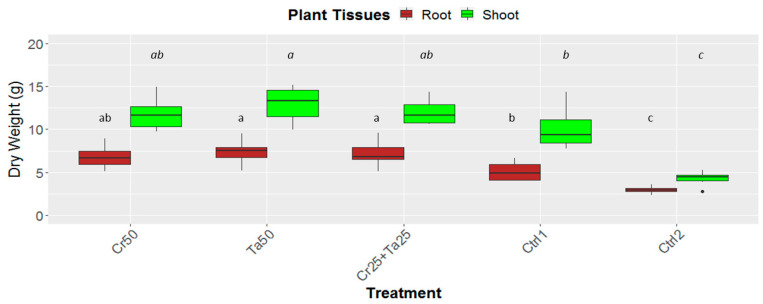
Effects of *Ta* and *Cr*, either applied individually or in consortium on the root and shoot dry weight of oil-palm seedlings at 5 months post-inoculation (MPI). Treatments abbreviations: Cr50—inoculated with 50 g of *C. rosea* AAB0114 (*Cr*); Ta50—inoculated with 50 g of *T. apiculatus* AAT0115 (*Ta*); Cr25+Ta25—inoculated with 25 g of *Cr* and 25 g of *Ta*; control 1—only the uninoculated blended maize was applied; and control 2—without the uninoculated blended maize. Means of the root and shoot dry weights were analyzed separately. Means with the same letters for the all five treatments the respective root and shoot dry weights were not significantly different with ANOVA–Fisher at *p* = 0.05.

**Figure 6 microorganisms-08-01138-f006:**
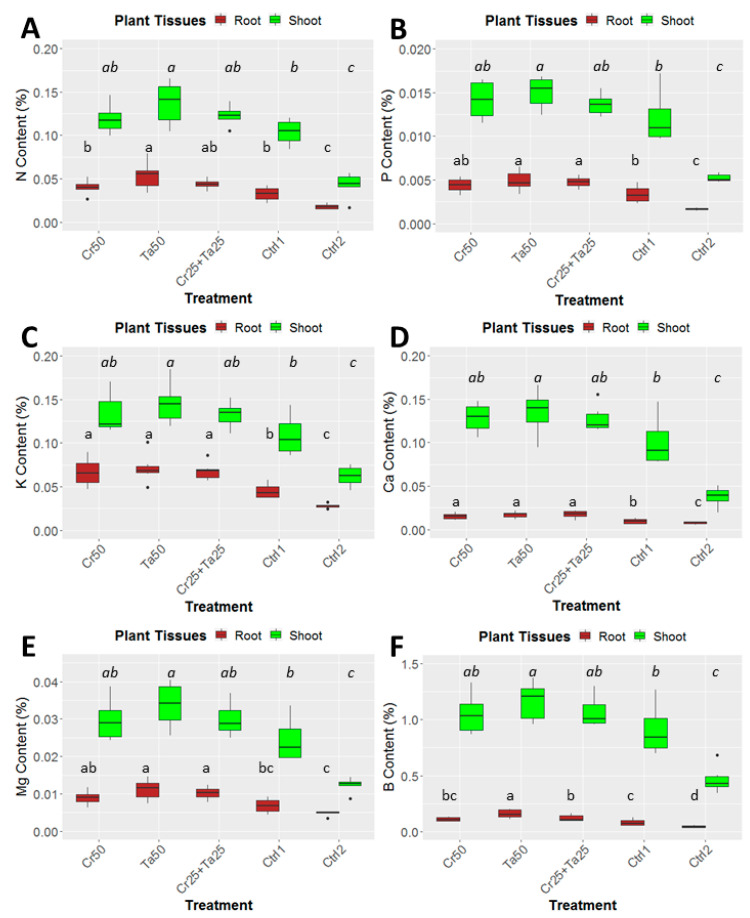
Effects of *Ta* and *Cr,* either applied individually or in consortium on the shoot and root nutrient contents in oil-palm seedlings at 5 months post-inoculation (MPI) and nutrients analyzed were (**A**) Nitrogen or N content, (**B**) phosphorus or P content, (**C**) potassium or K content, (**D**) magnesium or Mg content, (**E**) Calcium or Ca content and (**F**) Boron or B concentration. Treatments abbreviations: Cr50—inoculated with 50 g of *C. rosea* AAB0114 (*Cr*); Ta50—inoculated with 50 g of *T. apiculatus* AAT0115 (*Ta*); Cr25 + Ta25—inoculated with 25 g of *Cr* and 25 g of *Ta*; control 1—only the uninoculated blended maize was applied; and control 2—without the uninoculated blended maize. Means of the three different vegetative growth parameters (VGM) at two separate recording time points (3 and 5 MPI) were analyzed separately. Means with the same letters for the all five treatments within 3 or 5 MPI for the respective VGM parameters were not significantly different with ANOVA–Fisher at *p* = 0.05.

**Figure 7 microorganisms-08-01138-f007:**
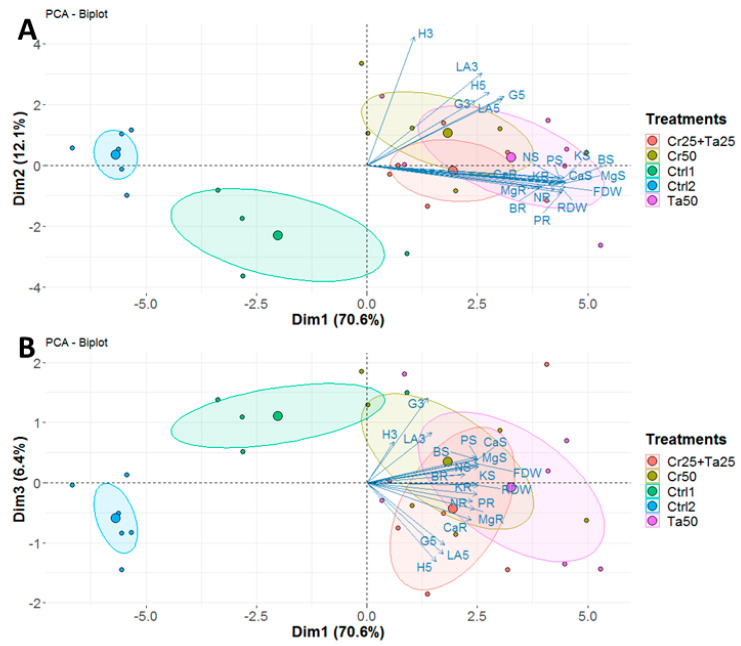
Principal component analysis (PCA) plots generated using (**A**) first and second dimensions and (**B**) first and third dimensions, based on various plant vegetative growth measurements and plant nutrient analyses for three treatments (Cr50, Ta50 and Cr25 + Ta25) and two controls (Ctrl 1 and Ctrl 2). Abbreviations: H, G and LA—height, girth and leaf area at 3- or 5-MPI; FDW—shoot dry weight; RDW—root dry weight; N, P, K, Ca, Mg and B–nitrogen, phosphorus, potassium, calcium, magnesium and boron for shoot (S) and root (R).

**Figure 8 microorganisms-08-01138-f008:**
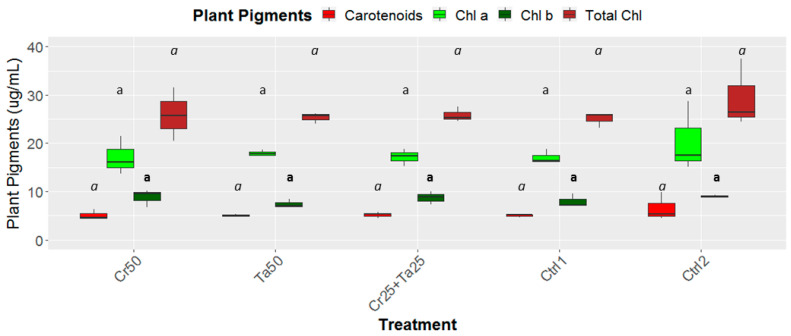
Effects of *Ta* and *Cr,* either applied individually or in consortium on the leaf chlorophyll as well as carotenoid contents and total chlorophylls for oil-palm seedlings at 5 months post-inoculation (MPI). Treatments abbreviations: Cr50—inoculated with 50 g of *C. rosea* AAB0114 (*Cr*); Ta50—inoculated with 50 g of *T. apiculatus* AAT0115 (*Ta*); Cr25+Ta25—inoculated with 25 g of *Cr* and 25 g of *Ta*; Control 1—only the uninoculated blended maize was applied; and Control 2—without the uninoculated blended maize. Means of the four different plant pigments, namely chlorophyll a (Chl a), chlorophyll b (Chl b), carotenoids and total chlorophylls (Total Chl) were analyzed separately. Means with the same letters for all five treatments for the respective plant pigments were not significantly different with ANOVA–Fisher at *P* = 0.05.

**Table 1 microorganisms-08-01138-t001:** Treatments and controls with their respective descriptions adopted for plant-growth-promoting experiment using AAB0114 (Cr) and AAT0115 (Ta) isolates.

Treatments	Descriptions
Cr50	Inoculation with 50 g of blended AAB0114 (Cr) inoculant
Ta50	Inoculation with 50 g of blended AAT0115 (Ta) inoculant
Cr25+Ta25	Inoculation with consortium of 25 g Cr and 25 g Ta inoculant
Ctrl 1 (control 1)	Application of uninoculated blended maize (50 g) only
Ctrl 2 (control 2)	Without uninoculated blended maize

**Table 2 microorganisms-08-01138-t002:** Reduction of linear mycelial growth of four *Ganoderma boninense* isolates by *Talaromyces apiculatus* AAT0115 and *Clonostachys rosea* AAB0114 on malt extract agar (MEA) 14 days after inoculation (DAI).

*Ganoderma boninense* Isolates	Pathogenicity Level ^§^	Treatment	Linear Mycelial Growth of *Ganoderma* Colony (mm) ^a^	
			Ta	Cr
*G. boniense* isolate G8	Low	control	44.17 (1.77) c	44.17 (1.77) c
		With Ta or Cr	20.47 (1.57) e	22.73 (1.10) e
*G. boninense* isolate G10	High	Control	53.30 (0.25) a	53.30 (0.25) a
		With Ta or Cr	26.53 (0.20) d	32.47 (0.87) d
*G. boninense* isolate G14	High	Control	53.13 (0.33) a	53.13 (0.33) a
		With Ta or Cr	25.47 (2.01) d	31.07 (2.11) d
*G. boninense* isolate G12	Moderate	Control	48.03 (1.36) b	48.03 (1.36) b
		With Ta or Cr	25.60 (0.84) d	24.73 (2.08) e

Abbreviations: Ta: *Talaromyces apiculatus* AAT0115 and Cr: *Clonostachys rosea* AAB0114. ^§^ Pathogenicity levels were based on disease scorings, namely disease incidence and disease severity index of oil-palm seedlings, reported in the previous study for the respective *G. boninense* isolates [32]. ^a^ Linear mycelial growth of *Ganoderma* colony (mm) challenged with or without (control) *T. apiculatus* and *C. rosea* in dual-culture assays on MEA were analyzed separately for the respective *Ta* and *Cr* columns; means of linear *Ganoderma* colony’s growth (mm) within individual *Ta* and *Cr* columns for all the four *G. boninense* isolates followed by the same letter are not significantly different at *p =* 0.05 after ANOVA–Fisher’s test. Numbers in the bracket are the standard errors for the respective means.

**Table 3 microorganisms-08-01138-t003:** Effects of Ta and Cr, either applied individually or in combination on the development of oil palm BSR by *Ganoderma boninense.*

Treatment ^a^	Disease Census							
	DI (%) on MPI ^b^			DSI (%) ^c^ on MPI ^b^		AUDPC (%) ^d^	DR (%) ^e^	
	3	4	5	3	4	5		
+G10–Cr–Ta	40	70	100	16.67 (7.86) a	40.00 (10.90) a	75.00 (7.14) a	54.17 a	–
+G10+Cr50	10	50	80	5.00 (5.00) ab	36.67 (14.00) a	61.67 (13.20) ab	35.84 ab	33.84
+G10+Ta50	30	60	80	16.67 (9.30) ab	35.00 (12.50) a	70.00 (11.90) a	51.67 a	4.62
+G10 + Cr25 + Ta25	0	20	60	0 b	11.67 (8.62) a	51.67 (14.20) ab	25.8 ab	52.37
+G10 + Cr50 + Ta50	10	20	50	5.00 (5.00) ab	18.33 (12.30) a	33.33 (12.90) b	21.67 b	60

^a^ Treatments: +G10-Cr–Ta—with RWB inoculated with *G. boninense* isolate G10 (G10) only with 50 g of uninoculated ground maize; +G10 + Cr50—with RWB inoculated with G10 and 50 g of *C. rosea* AAB0114 (Cr); +G10 + Ta50—with RWB inoculated with G10 and 50 g of *T. apiculatus* AAT0115 (Ta); +G10 + Cr25 + Ta25—with RWB inoculated with G10, 25 g of Cr and 25 g of Ta; and +G10 + Cr50 + Ta50—with RWB inoculated with G10, 50 g of Cr and 50 g of Ta. ^b^ MPI refers to months post-inoculation. ^c^ Means of DSI at three separate months post-inoculation (3, 4 and 5 MPI) were analyzed separately. Means within each column of MPI followed by the same letter are not significantly different at *p* = 0.05 after ANOVA–Fisher’s test. ^d^ AUDPC refers to area under disease progress curve. Means within the column of AUDPC followed by the same letter are not significantly different at *p* = 0.05 after ANOVA–Fisher’s test. ^e^ DR refers to percent disease reduction in percent of AUDPC. Numbers presented in the figures are untransformed means.

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
