# Peer review of "Biocontrol and Plant-Growth-Promoting Traits of Talaromyces apiculatus and Clonostachys rosea Consortium against Ganoderma Basal Stem Rot Disease of Oil Palm"

_microorganisms, 2020, doi:10.3390/microorganisms8081138_

Round 1

Reviewer 1 Report

This paper describes a series of experiments assessing two fungal strains for control of basal stem rot on oil palm. The paper is generally well written and follows a common screening route, via identification by ITS sequencing, in vitro assessment of beneficial characters (pathogen inhibition, fungi-pathogen interactions) followed by in planta tests for plant growth promotion (including measurement of nutrient and chlorophyll content) and disease control. Experimental design and analysis is appropriate.

The paper describes the isolation and assessment of only 2 fungal isolates. Were these the only two isolates that were recovered from RWB or diseased palms? Some background on isolation should be given or a reference. Was there a screening system to initially identify these strains?

Pathogen inhibition on agar is a common method, and is good they used a number of G. boniennes isolates of different pathogenicities for comparison. I don’t think statistical analysis of data transformed to % inhibition is appropriate. It would be preferred to use the actual data on fungal growth (fungi diameter, size of inhibition zone) and include control to discount effects on transformation to PIRG%.

In biological control, the important data is control of the target disease on plant, however, in this case there is only 5 lines on the results of the disease control experiment. The authors state the experiment was repeated twice but only one set of data is presented in Table 3, with no indication if this was combined data or from only one of the experiments. As this is the crux of the research, details of both experiments should be presented so readers can assess variability. Also, there is no data on control without the pathogen G10, this should be included for comparison. Also, plant growth data should also be presented as further evidence of benefit from Ta and Cr. As is, the reader cannot determine if inoculation with G10 had any impact on plant growth. Another problem is the authors overstate the case for disease control as from Table 3, only the combination of Cr50+Ta50 is significantly different from the no CR-Ta control at 5 months, and there is no statistical data on disease index and AUDPC, this should be included.

The claim to improve chlorophyll and carotenoid contents is not correct (L445). The data presented in Figure 8 shows no difference between treatments, and means for control Ctr2 are higher than Ta and Cr treatment. There may however be an increase in total amounts due to increased leaf area but data for this is not presented.

Minor issues

L121, L474, L574 L592, in vitro should be in italics

L150 formula for PIRG should be stated

152 Brief description of conditions should be given

L214. Formulas and method for determining DI and DSI should be given in text as well as reference.

Figure 1. strain numbers should be prefixed with species name or abbreviation (Ta, Cr) for clarity.

Figures 2 and 3. Text in dendrogram is not clear and difficult to determine where the 2 strains in bold are located.

L254. C and D should be A and B

Figures 4-8. Axis labels are not clear enough, increase font size

Figure 5. Why was violin plots used instead of the box and whisker plots used for other graphs? I think better to be consistent.

L279-280. Change “on the contrary…” to “In contrast…”

L312. Change “correlated’ to “associated” might be more accurate.

L660, L682, L707, L710, Species names should be italics

Reference 74 is in title case, change to sentence case

L150, 191, L217 and other places. author name should be included before reference number, e.g. …proposed by Ting et al. [38]. And …adopted by Sharifuddin et al. [45].

Reviewer 2 Report

This paper investigates the performance of two potential biocontrol agents, namely Clonostachys rosea (Cr or AAB0114) and Talaromyces apiculatus (Ta or AAT0115), against Ganoderma boninense, a basidiomycetes which causes basal stem rot disease. The authors also investigated their contribution to the plant growth and showed that they may promote performance. Results from in vitro studies showed that both Ta and Cr caused an inhibition of G. boninense mycelia growth either individually or in combination and thus they both show potential as BSR biocontrol agents.

The article is of interest and the results are presented well. However, there are several questions and comments raised which render this manuscript not suitable for publication at its current form. The main questions/comments raised are:

  1. In the molecular identification of Cr and Ta, the authors do not explain if during the phylogenetic analyses have constructed the trees produced by ML with the use of a model (e.g. Kimura 2-parameters, GTR, etc) and how have they selected the taxa which comprise their matrices. I was expecting to read these details at M&M. Additionally, I realise for Ta the need for a second molecular marker besides ITS but I do not see why the authors have chosen benA and not any other molecular marker. 
  2.   While not extremely familiar with the inhibition tests, I see that p-values vary greatly (e.g. l. 273: 0.08-0.83). How do the authors explain this? Can we accept values larger than 0.05 ot at least 0.1?
  3. In M&M 2.4 but also in the Results the authors state that for single treatments they have used 50g but for the combined bioassays they have used 25+25g (and not 50+50g). Based on which strategy have they changed the quantities applied? Is there any reference on which they are based on?
  4. Finally, I am really puzzled why the authors present in their discussion all secondary metabolites produced by these two mycoparasites, when their work does not include any such analysis. I realise the importance of defining the metabolites which may play crucial role in the mechanism of mycoparasitism but at the moment as it is presented (l. 402-425), this data is not well incorporated within the discussion.

Minor comments:

About the molecular analyses:

L. 131 & 136: while the primers used are universal beyond any doubt,  it is advisable to also provide them as sequences. for the readers who are not familiar with them, as otherwise they have to retrieve the specified references.

L. 238-239: percentages present sequence identity? or query coverage? please explain it in the text

L. 241 & 247-248: why do the authors  need to separate AAT0115 from T. aculeatus, if it is already separated according to line 241?

Other minor points:

I think that references in the text should be in another format. Fo instance,  " as proposed by Kok et al., 2013 [no of ref.]" and not as: "as said by [no of ref]" - check please with the guidelines of the journal on this.

Moreover, the authors have to go through the whole ms and check the syntax of the ms as there are many mistakes wwhich make it difficult to comprehend. Several examples are provided below:

Line 35: re-phrase "not negatively unaffected". Does that mean "positively affected" or "not affected at all"?

Line 60: needs refs

Line 71" please replace "illustrated" with "showed" or "presented"

Line 73: please re-phrase as it does not make sense the need to "minimize the variation of BCAs effectiveness"

Line 74: "capable of" instead of "in"

Line 94 "was proposed it to be ..." replace with "... and they have suggested that it is..."

103-105: re-phrase

Line 129: for the 1st time in the text explain which medium MEA is

Line 132: verb is missing

Line 135: why only for aat0115-please explain

Line 144: wrong usage of verb "to challenge"

Line 154-why 50% polyethylene shading net?

Line 257:replace "incorporated" with "used". Why have these species were chosen as outgroups?

Line 282-283: "possible…was observed". What does possible mean? was it or wasn’t it observed?

Line 377-379: rephrase

Line 382-383: And what does the reader realises from that?

Line 400: "the a few": does not make sense

Line 400-401: rephrase, the sentence does not have a clear meaning

Line 401: "combative"?

Line 403: "armed"? do the authors mean: "... as it carries in its genome gene clusters which encode..."

 Line 408:gene in italics

Line 418: erase "too"

Line 426-427: rephrase

Line 478-483: rephrase
